# Machine learning-based real-time object locator/ evaluator for cryo-EM data collection

Koji Yonekura [1,2,3✉], Saori Maki-Yonekura[1], Hisashi Naitow [1], Tasuku Hamaguchi [1] & Kiyofumi Takaba[1]

In cryo-electron microscopy (cryo-EM) data collection, locating a target object is error-prone. Here, we present a machine learning-based approach with a real-time object locator named yoneoLocr using YOLO, a well-known object detection system. Implementation shows its effectiveness in rapidly and precisely locating carbon holes in single particle cryo-EM and in locating crystals and evaluating electron diffraction (ED) patterns in automated cryo-electron crystallography (cryo-EX) data collection. The proposed approach will advance high-throughput and accurate data collection of images and diffraction patterns with minimal human operation.

[1] Biostructural Mechanism Laboratory, RIKEN SPring-8 Center, Sayo, Hyogo, Japan. [2] Institute of Multidisciplinary Research for Advanced Materials, Tohoku University, Sendai, Japan. [3] Advanced Electron Microscope Development Unit, RIKEN-JEOL Collaboration Center, RIKEN Baton Zone Program, Sayo, Hyogo, Japan. ✉email: yone@spring8.or.jp

ryo-EM work can extend over hours or days during which thousands of image stacks are collected for high-resolution single particle reconstructions. Automated data collection with minimal human-supervision would be advantageous. A number of computer programs have been developed to fulfill this (e.g. refs. [1–3]), and are widely used in laboratories and shared facilities. Automation is also becoming increasingly important in electron 3D crystallography/3D ED/microED, where acquisition of rotational diffraction data from many small crystals is often needed (e.g. refs. [4,5]).

The operator's tasks in single particle analysis (SPA) data collection typically comprise: alignment of the microscope; selection of grid squares and carbon holes containing sample molecules from low magnification images; and start data acquisition. Data acquisition requires expediting search, focus, and record modes. In search mode, a target carbon hole with a diameter of 1–2 μm and filled with amorphous ice is positioned at intermediate magnification. This maneuver is error-prone due to discrepancies in stage positions between low and intermediate magnifications and a poor positional reproducibility or mechanical instability of the specimen stage. The stage rarely goes to an exact pre-registered position within an allowable error range, typically, a few hundred nano-meters. Precise adjustment of the stage position in search mode is critical and yet often frustrating. Stage alignment to a hole is done by cross-correlating a test image at 1,000× to 10,000× magnification and a reference, an ideal hole image at the same magnification or a corresponding image at lower magnification, prepared beforehand. However, this step can be problematic, particularly when poor signals in thick ice, incorrect or broad correlation peaks of partially recorded multiple holes, and contamination. Accurate positioning is being more critical, as many image stacks are acquired from nearby areas without repositioning the stage for high-throughput data collection[6].

The same transposition problem arises in automated cryo-EX data collection. Since the appearances of crystals differ from crystal to crystal, it is more challenging than for hole detection. Here we introduce a new approach based-on machine-learning for real-time object locator and have applied it to automated cryo-SPA and EX data collection. The program can also evaluate the quality of diffraction patterns worthwhile to proceed to data collection.

## Results and discussion

**Cryo-SPA data acquisition.** Modern schemes take 9–25 or more image stacks from holes clustered around a central hole (Fig. 1a)

and/or even more stacks for one hole (Fig. 1b) through changing deflector coils once the stage is moved to a new position[6]. No additional image used for alignment is acquired for each surrounding hole. This approach is indeed very effective to speed up data collection, but, when the alignment fails, it can be disastrous, yielding much useless data, wasting time and data storage space. To avoid this, EM operators can try higher defocus values, longer exposure times, larger frame binning and, if applicable, filtration of inelastically scattered electrons[7] for gain in image visibility. However, these interventions require operator input, is time-consuming, and can be ineffective in difficult cases. Also, during unsupervised data acquisition multiple trials for positioning one hole may be repeated, resulting in possible radiation damage.

Instead of relying on cross correlations, we have developed a new algorithm incorporating machine-learning (Fig. 2) and the well-known real-time object detection system YOLO (You Only Look Once)[8]. Machine-learning has already been introduced in the cryo-EM field to solve various problems in data analyses such as particle picking (e.g. refs. [9–11]), denoising (e.g. ref. [12]), analysis of structure variations (e.g. ref. [13]) and so on. However, it has not, or has hardly, been used in the control and supervision of data collection to our knowledge, while YOLO was originally designed for real-time object detection. Our software named yoneoLocr (You Only Navigate EM Once to LOCate in Real time) is based on the latest release of YOLO version 5 (YOLOv5; https://github.com/ultralytics/yolov5), and can detect objects in less than 0.1 s using trained weights.

Implementation required using cryo-EM images containing carbon holes recorded at a nominal magnification of 8,000× on a Gatan K3 direct electron detection (DED) camera with a JEOL CRYO ARM 300 electron microscope[14,15]. This magnification is generally used for stage alignment in search mode of SPA data collection, but here the in-column type energy slit is retracted due to a large cut in the view. We then enclosed the carbon holes in these images with a box and annotated them as "hole". The annotated images were trained with a network model of YOLOv5s on 4× GPU cards installed on a Linux workstation. Data statistics for training are summarized in Table 1. Hole detection was excellent (Fig. 3). Unsurprisingly, YOLOv5 outperformed YOLOv3, and the simplest model YOLOv5s showed a superb performance for hole detection. The trained weight succeeded in the detection of holes recorded at different magnifications (×2.5k) and holes in gold grids (Fig. 4).

We made a python program named yoneoLocrWatch.py (Fig. 2) to monitor updates in a predefined directory of a K3 control Windows PC equipped with two GPU cards. One GPU

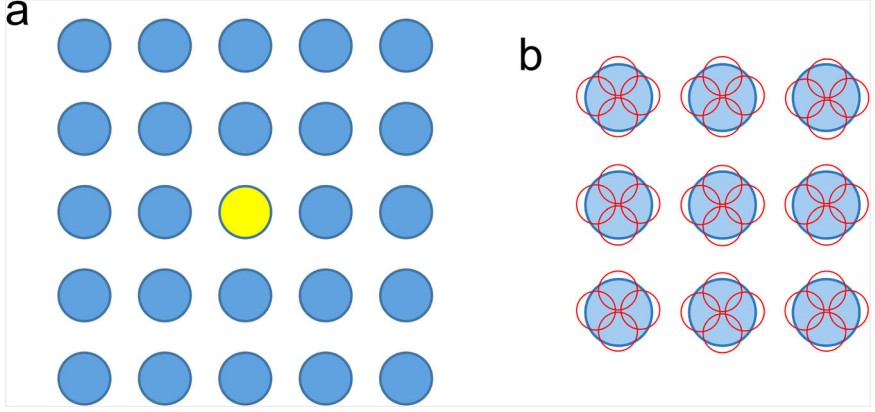

**Fig. 1 Schematic diagram of data taking for SPA through image shifts. a** Stage shift is only used to locate the central hole in yellow, and image shifts are applied for data taking from other 24 holes in this example. **b** Multiple images stacks, 4 in this example, are recorded from different areas shown in small red circles in one hole. Pre-check of the surrounding holes is unsuitable for fast collection of images stacks from each hole.

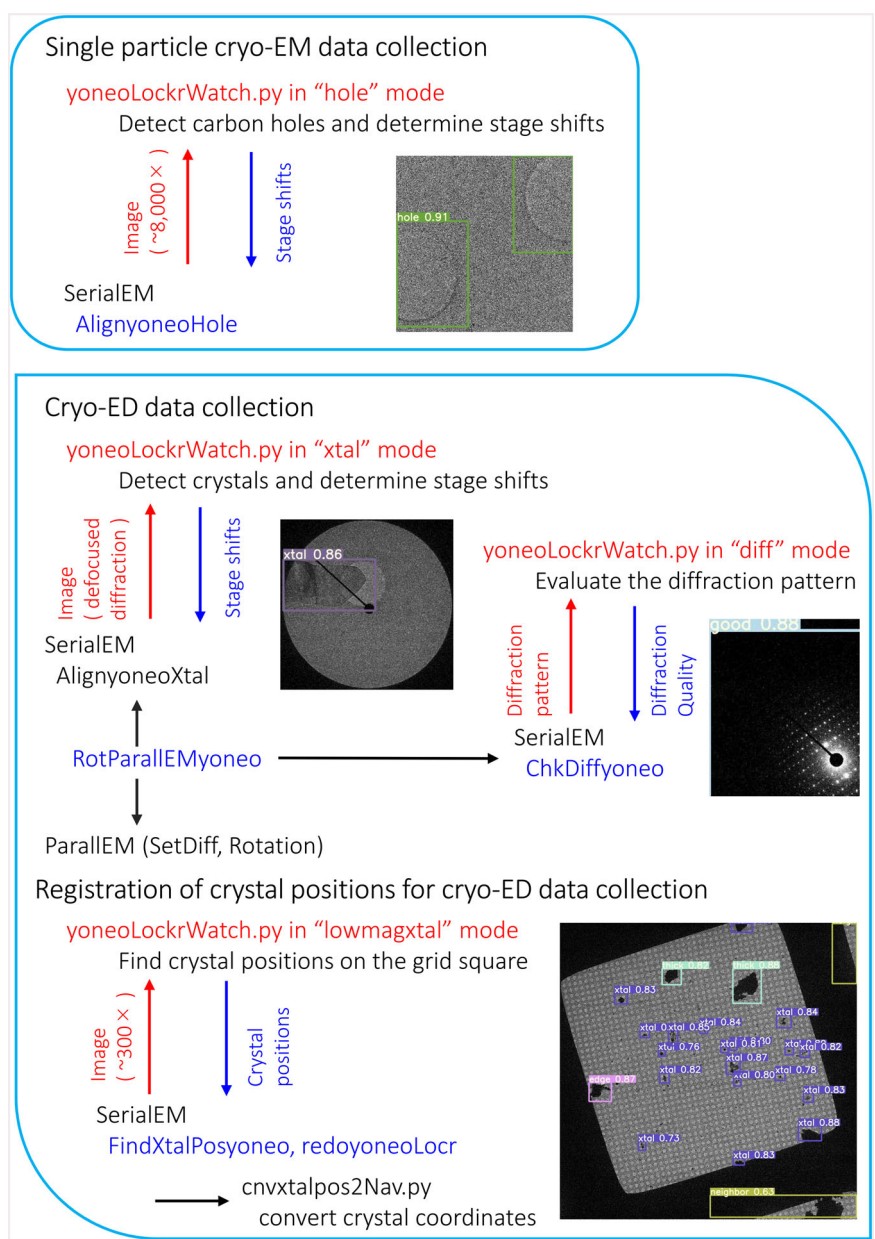

**Fig. 2 Schematic diagram for positioning the target and checking the quality of diffraction patterns by yoneoLocrWatch.py, SerialEM scripts and associated programs.** The main program yoneoLocrWatch.py runs in 4 distinct modes for stage alignment for a carbon hole (hole) and a crystal (xtal), for evaluation of diffraction patterns (diff) and for detection of crystals on low magnification images and registration of crystal positions (lowmagxtal). The SerialEM scripts call and control yoneoLocrWatch.py and other associated programs including ParallEM.

**Table 1 Data statistics for training of images and diffraction patterns with YOLOv5.**

| Mode | hole | xtal | diff | lowmagxtal |
|---|---|---|---|---|
| Numbers of original images | 300 | 331 | 751 | 33 |
| Resized dimension of images | 800 × 800 | 800 × 800 | 1024 × 1024 | 800 × 800 |
| Model | YOLOv5s | YOLOv5x | YOLOv5x | YOLOv5x |
| Training time (h) | ~1.9 | ~4 | ~20 | ~1 |

card may be enough as the computation and memory loads are low during running of the object detection routine and at other times the program is idle. Once a text file in the directory is renewed, the program reads an image indicated in the text file and provides the best directions for stage shifts based on confidence level, hole size, and distance from the center of the image. Hole detection typically takes ~0.06 s. We made a SerialEM script, AlignyoneoHole, for incorporating this hole detection step into automated data acquisition (Fig. 2). The script takes a new image, updates the directory, reads a log, and shifts

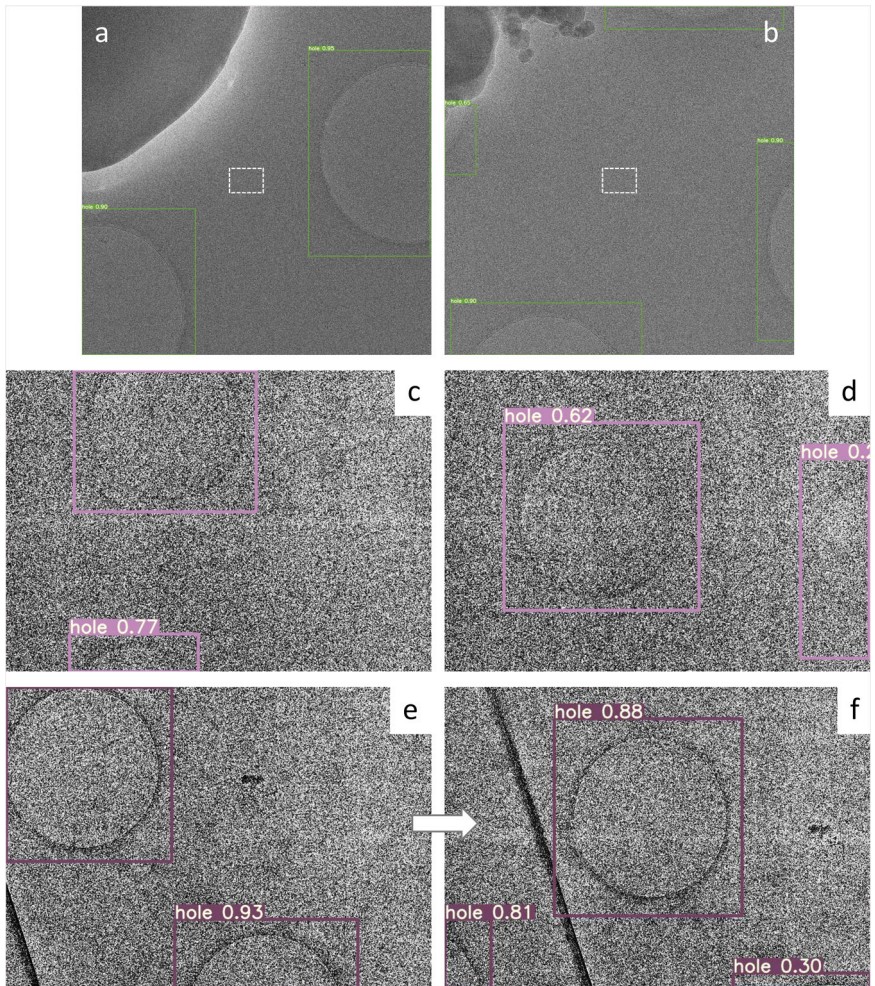

**Fig. 3 Detection of carbon holes. a, b** Failures of stage positioning were done by cross-correlation with a reference image using SerialEM due to excessive contamination in dark blobs and to partially recorded multiple holes. Center small rectangles (white broken lines) indicate a rough position and dimension of acquired movie stacks on the K3 camera at a nominal magnification of 100,000×. The horizontal size of the rectangles corresponds to ~0.28 μm. Thus, each of these alignments produced 25 useless image stacks, as image shifts were applied over 5 × 5 neighboring holes for data collection at that time. YOLOv5 with a weight trained in this study detected all holes (boxes) correctly. **c, d** Low-contrast images of holes covered with a thin carbon layer and with thick ice. The trained weight still succeeded in hole detection (boxes). **e** The first image taken in search mode. **f** The second image after stage adjustment by yoneoLocrWatch.py running in hole mode and AlignyoneoHole.

the stage if needed. This approach never failed in stage alignment, even for difficult samples with large contamination and less visibility of thick ice or carbon film over holes, as long as holes were recognizable by the human eye (Fig. 3a–d), in notable contrast to the cross-correlation method (Fig. 3a, b). In one dataset collected using yoneoLocr from a holey carbon grid covered with a thin carbon layer, ~91% of images were correctly aligned within two trials (218 of 240) and the remaining ~9% were aligned by the third trial (239 of 240). The success rate within second trials reached ~93% for typical samples (528 of 565) on standard holey carbon grids or gold-sputtered holey carbon grids[14]. The final success rate was also increased from ~78 to ~100% with running yoneoLocr for hole detection (Table 2). Of course, this is a qualitative measure, as the performance of locating holes varies depending on each sample condition. Obviously, there are additional advantages - ~5–50-fold less exposure time (decreased from 0.5–2 s to 0.03–0.2 s for taking one search image and requiring less repeats till positional error is small enough), therefore less radiation damage, and no or less defocusing, therefore minimal objective lens changes and more stable data collection. A typical job for hole detection and stage alignment is shown in Supplementary Movie 1.

**Cryo-EX data acquisition**. There have been several reports and programs for automated collection of rotational ED data (e.g. refs. [16,17]). Our group also developed and reported a scheme that combines SerialEM for positioning of sample crystals and ParallEM for controlling data acquisition[18]. This protocol needs manual positioning of crystals and registration of the coordinates in defocused diffraction (search) mode before unsupervised data acquisition is started in focused diffraction (data-taking) mode. Thus the same transposition problem is seen in SPA data collection arises. Since the appearances of crystals differ from crystal to crystal, it is more challenging than for hole detection, as mentioned. The original protocol does not include re-positioning to target crystals during data collection and is more severely affected by positional reproducibility.

YOLO was used again for training with various crystal images of proteins[19], polypeptides, organic molecules[5], and semiconductor materials[4] recorded in defocused diffraction mode on different cameras, two scintillator-coupled CMOS cameras, Gatan OneView[19] and TVIPS XF416[5,15], and one DED camera Direct Electron DE64[4]. As samples often contain ice crystals of characteristic hexagonal shape and yielding strong hexagonal diffraction patterns, crystal images were annotated as "xtal" or

"ice" according to appearance. We also trained YOLOv5 with ED patterns of crystals acquired in data-taking mode. Diffraction patterns were visually assigned as "good", "soso", "bad", "no" and "ice" depending on the quality of the patterns within a single box covering most of the whole image. The worst level "no" had patterns with no diffraction spots. The most accurate model (YOLOv5x) was adopted for both crystal images and diffraction patterns.

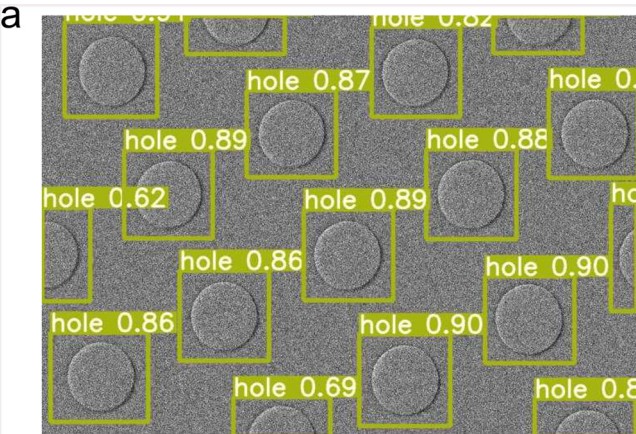

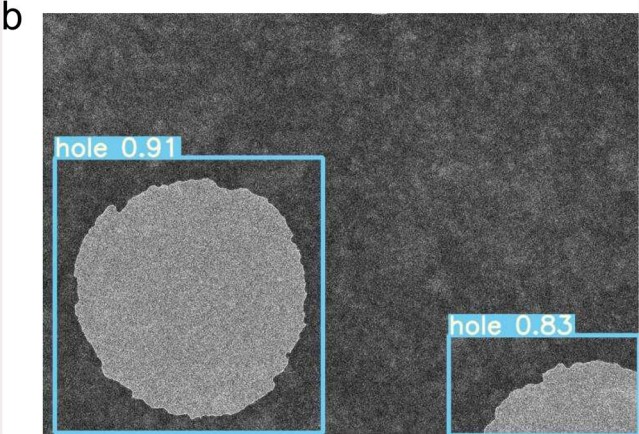

**Fig. 4 Hole detection in different conditions. a** Carbon holes taken at 2,500×. The hole size is ~1.5 μm. **b** Holes on a gold grid (Quantifoil UltrAuFoil R 1.2/1.3) taken at 8,000×. Histogram equalization was not used for better detection of the gold holes. The hole size is ~1.2 μm.

Runs, incorporating the trained weights, were implemented in DE64 and XF416 camera control PCs equipped with GPU cards. Two sessions of yoneoLocrWatch.py were launched simultaneously in two different modes, "xtal" for locating a crystal and "diff" for evaluating the diffraction pattern (Fig. 5). Two SerialEM scripts, AlignyoneoXtal and ChkDiffyoneo, were made for incorporation of these operations into script RotParallEMyoneo that calls up ParallEM for sequential data collection of rotational diffraction patterns in a queue list in the Navigator window of SerialEM (Fig. 2). SetDiff in ParallEM switches between defocused (search) and focused (data-taking) modes by changing only the intermediate lens 1 (IL1) value. During data collection, it never goes back to imaging nor low magnification modes. Thus, possible hysteresis from lens changes is minimal and the beam setting is stable even over operations taking several days, which could include overnight breaks with the beam off and multiple flashing cycles for refreshing the cold-field emission gun in the CRYO ARM 300 microscope[14,15,18]. SetDiff also controls insertion and retraction of the selected aperture and energy slit[15,18].

Thus, identifying suitable crystals and positioning one to center view are possible in defocused diffraction mode, as is done for carbon holes in the SPA data collection scheme (Fig. 5a, b). This step mostly performed well, but, different from regular hole patterns, moving to the best crystal sometimes failed when the crystals are, for example, crowded or too thick. Thus, a prior quality check of untilted still diffraction patterns would be helpful, as acquisition of one rotational data set takes 2–3 min and rotational frames fill large disk space. This can be done in data-taking mode with a short exposure of ~0.1 s (Fig. 5c; Supplementary Fig. 1). It works particularly well for protein and organic semiconductor crystals, which yield many diffraction spots. Data taking with other crystals may succeed at the low assessment levels "soso" or even "bad" (Supplementary Fig. 1). The worst level "no" assigned for no diffraction spots was always easily determined for all samples tested in this study (Supplementary Fig. 1c). Once the diffraction pattern is judged to be above a given level, RotParallEMyoneo calls up Rotations in ParallEM to start rotational data acquisition (Fig. 2). A typical job for crystal detection, stage alignment, and diffraction evaluation is shown in Supplementary Movie 2.

Finally, we annotated crystals in single grid squares in low-magnification images (250–400×) and trained with YOLOv5x again. We categorized crystals as "xtal" for candidates for data collection, "thick" for dense samples, and "edge" for those at the periphery of the grid square roughly normal to the rotation axis. Data collection from crystals at the edge may be blocked by the grid bar at high tilt angles. We also annotated neighboring grid

**Table 2 Typical data statistics for locating carbon holes in SPA data collection with and without running yoneoLocr.**

**Without yoneoLocr**

| Date | Grid type | Total image number* | Success | Fail | Success rate (%) |
|---|---|---|---|---|---|
| 2020.12.25–2020.12.27 | Gold sputtered | 16,814 | 13,214 | 3600 | 78.6 |
| 2021.1.27–2021.1.29 | Covered with thin carbon film | 4223 | 3150 | 1073 | 74.6 |
| 2021.2.8–2021.2.9 | Gold sputtered | 3975 | 3250 | 725 | 81.8 |
| Total | | 25,012 | 19,614 | 5398 | 78.4 |

**Running yoneoLocr in hole mode**

| Date | Grid type | Total image number* | Success | Fail | Success rate (%) |
|---|---|---|---|---|---|
| 2021.2.18–2021.2.19 | Covered with thin carbon film | 5927 | 5927 | 0 | 100 |
| 2021.3.8–2021.3.9 | Covered with thin carbon film | 6749 | 6724 | 25 | 99.6 |
| 2021.3.18–2021.3.23 | Gold sputtered | 23,663 | 23,663 | 0 | 100 |
| Total | | 36,339 | 35,684 | 36,314 | 99.9 |

* Numbers of center holes adjusted by stage shift are ~1/25 of the total image numbers.

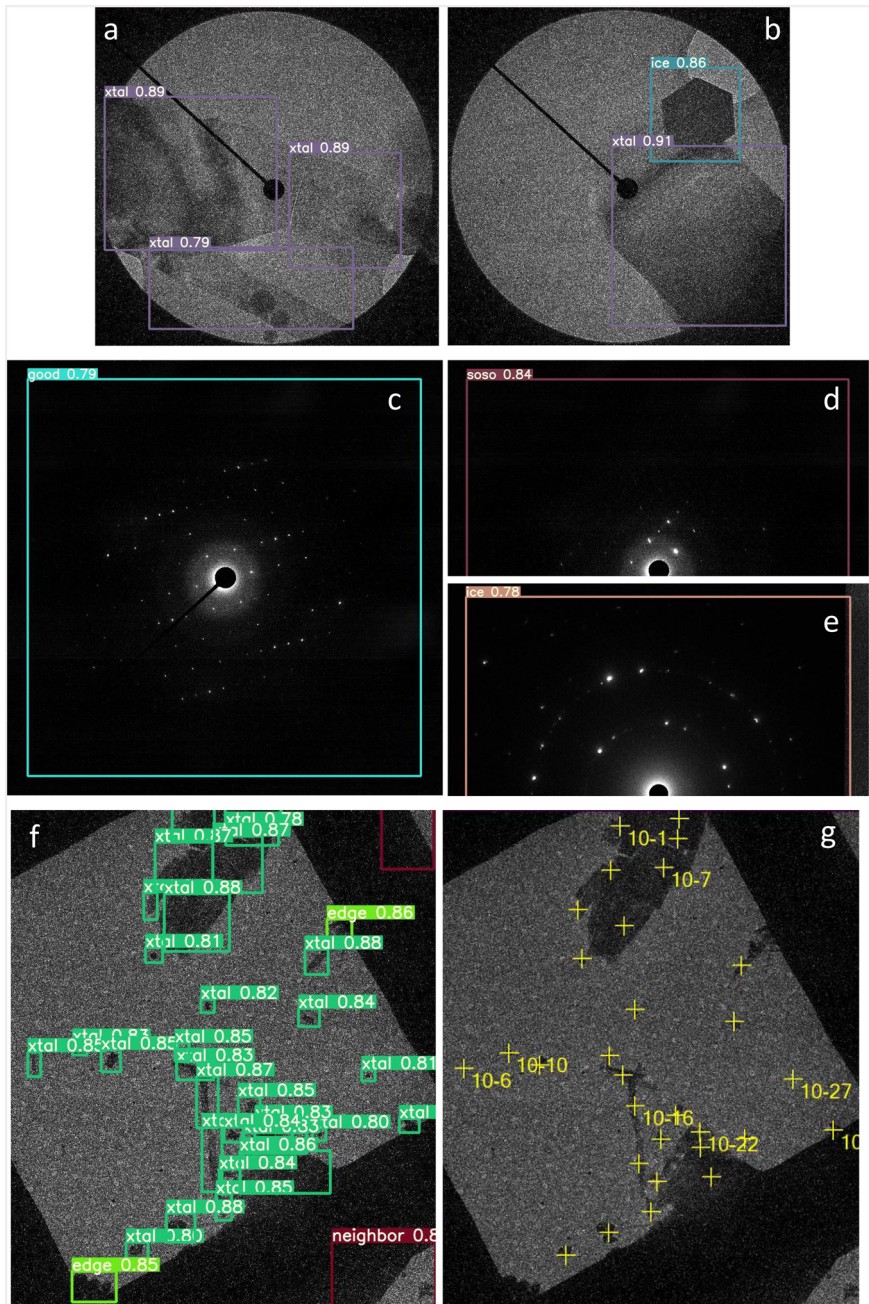

**Fig. 5 Detection of crystals and check of diffraction patterns. a**, **b** Detection of crystals in defocused diffraction mode by yoneoLocrWatch.py in xtal mode. A hexagonal ice crystal was identified in b. The beam size is ~5 μm in a and b. **c**, **d**, **e** Evaluation of diffraction patterns by the same python program in diff mode. A diffraction pattern from an ice crystal was identified in **e**. **f** Detection of crystals taken in a low-magnification (300×) image in lowmagxtal mode. Crystals at the square edge were also identified, and crystals were not selected from neighboring squares. **g**. Registration of crystal positions to a queue list in SerialEM by FindXtalPosyoneo and cnvxtalpos2Nav.py. The crystals at the edge were automatically excluded in this case. The square size is 85 × 85 μm².

squares as "neighbor" to exclude crystals on other squares. Thus, the approximate positions of crystals can be obtained by yoneoLocrWatch.py running in the new mode "lowmagxtal", and SerialEM script, FindXtalPosyoneo, which calls up a python script, cnvxtalpos2Nav.py, to convert crystal coordinates, and adds the positions to a queue list (Figs. 2, 5f, g). This step can be retried with a new confidence threshold level for crystal selection by redoyoneoLocr without retaking images (Supplementary Movie 3). Users may also edit the queue list. Now, registration of suitable

crystals and unsupervised rotational data collection are almost fully automated by combining the three implementations in yoneoLocr and associated programs above. Automatic positioning to crystals can be implemented in several published programs, which set certain threshold pixel values to define crystal areas[16,20], but they cannot discriminate ice crystals, nor avoid thicker ones.

The success rate was increased from 52.8% to 100% (Table 3) when an assessment level of "good" + "soso" for diffraction patterns was used. Again this is a qualitative measure.

**Table 3 Typical data statistics for automated cryo-EX data collection with and without running yoneoLocr.**

| Date | Without yoneoLocr | | | | |
| | Manual selection (h) | Collected diffraction data sets | Data quality | | Success rate ("good" + "soso") (%) |
| | | | bad | no | |
|---|---|---|---|---|---|
| 2020.10.14–2020.10.15 | > ~8 | 80 | 17 | 18 | 56.3 |
| 2021.2.25–2021.2.26 | > ~8 | 81 | 27 | 14 | 49.4 |
| Total | | 161 | 44 | 32 | 52.8 |

| Date | Running yoneoLocr in lowmagxtal, xtal, and diff modes | | | | |
| | Automated selection (min) | Collected diffraction data sets | Data quality | | Success rate ("good" + "soso") (%) |
| | | | bad | no | |
|---|---|---|---|---|---|
| 2021.6.08 | ~10 | 188 | 0 | 0 | 100 |

Nevertheless, our approach can remarkably reduce the time and labor, whereas manual selection/registration of good-diffracting crystals usually takes a long time.

## Conclusions

In summary, we have developed a real-time object locator/evaluator yoneoLocr for cryo-EM data collection based on machine-learning, which obviates development of special algorithms for each target. Application of the software with SerialEM scripts is effective in efficient unsupervised data collection for SPA and electron 3D crystallography. Object detection is very fast, and can be done with small CPU and GPU loads and minimal memory. There is no delay in running SerialEM and ParallEM, usage is simple, and the routine can be called from other software. Locating carbon holes is extremely precise, with short exposure time, and is even effective in hitherto difficult cases—greatly benefiting SPA data collection. We could also train the detection system on various images such as lacey holes, empty ones, and holes with thick ice and contaminations, as proposed previously[21], but it would be less useful for automated data collection through image shifts, as this scheme always uses a regular array of holes and skips pre-check of all surrounding holes around the center one for speeding the whole process up. Image selection could be done just after image acquisition by using other software such as Warp[11] and cryoSPARC Live[22]. Implementation for automated cryo-EX data collection also performs well, but may need further training with more ED data. The program is also able to re-tune the weight with new data during data collection, which is a function implemented in YOLOv5. We have yet to thoroughly test this feature.

## Methods

YOLOv5 (https://github.com/ultralytics/yolov5) was installed under python 3.8 environment created by Anaconda in a linux work station equipped with 4× NVIDIA GeForce Titan X GPU cards. This version of YOLO uses PyTorch for deep learning together with CUDA, cuDNN, and so on. A real-time object locator yoneoLocrWatch.py developed in this study was set up under the same environment created by Miniconda or Anaconda on camera control PCs operated by Microsoft Windows Server 2012R2 for K3, and Windows 10 for DE64 and XF416 cameras. The program and associated scripts were placed in C:\ProgramData\ in the PCs. The K3 control PC is equipped with two GPU cards, Quadro K2200 and P6000, the DE64 PC with two Quadro RTX6000s and the XF416 PC with a Quadro K420 and a P4000. PyTorch in our configuration did not support the old Kepler-architecture K420 card. ImageMagick was used for pretreatment of images in yoneoLocrWatch.py. Details of the installation and command line options are presented in Supplementary method.

Cryo-EM images and ED patterns in jpeg format were annotated with GUI software labelImg (https://github.com/tzutalin/labelImg), and the format of coordinates and class names for annotated objects were converted to the YOLO format using convert2Yolo (https://github.com/ssaru/convert2Yolo). All training was done on the linux workstation using train.py included in YOLOv5. Detection test was performed with detect.py.

For hole images, several variations of original images were created by applying binning of images and/or histogram equalization using convert of ImageMagick. Training for the images took ~1.9 h using a network model YOLOv5s with an image size of 800 × 800. The most accurate model (YOLOv5x) was used for training of crystal images taken in search mode and diffraction patterns in data-taking mode. It took ~4 h with an image size of 800 × 800 and ~20 h with an image size of 1024 × 1024. Training of crystals taken in low magnification images was done using the YOLOv5x model. Each image contained many crystal images (Fig. 5f and g), which reached a good weight from a relatively smaller number of input images. Training statistics are shown in Table 1 and training metrics in "confusion_matrix.png" and "results.png" generated by YOLOv5 are given in Supplementary Fig. 2. The confusion matrixes suggest that misassignment may occur for some classes such as "ice" in xtal, "soso" in diff and "edge" in lowmagxtal modes, probably due to small numbers of training objects in the "ice" and "edge" classes and difficult cases in the "soso" class. Of course, the metrics strongly depend on training and validation datasets. Detection and classification of these objects were correct with the trained weights here in usual cases, but further training should be done with more data and proper labels.

For control and unsupervised data acquisition, yoneoLocrWatch.py keeps watching updates in a predefined directory, WatchHole\ in hole mode, WatchXtal\ in xtal mode, WatchDiff\ in diff mode and WatchLowmagXtal\ in lowmagxtal mode. Once a text file in these directories is renewed, the program reads an image indicated by the text file. Running in hole mode can enhance the image contrast by applying binning and histogram equalization. Then, it detects holes and gives the best answer for stage shifts in a log file based on the confidence level, the hole size, and distance from the center of the image. Running in other modes also gives the results of detection or evaluation in a log file.

SerialEM scripts AlignyoneoHole, AlignyoneoXtal, ChkDiffyoneo, and FindXtalyoneo take images, put a text file containing the location and name of a newly recorded image in the directories yoneoLocrWatch.py is watching, and obtain alignment parameters or quality check from a log file of yoneoLocrWatch.py. Then, AlignyoneoHole and AlignyoneoXtal move the stage if needed. As the pixel size cannot be defined in diffraction mode, we spread the beam to the detector edge and supply the beam size (~5 or ~7 μm) to AlignyoneoXtal as a scale for stage shifts. ChkDiffyoneo determines whether the corresponding crystal is worth rotational data collection or not. FindXtalPosyoneo finds crystals in low magnification images and adds the crystal position in a queue list in the Navigation window of SerialEM. This crystal detection can be retried with a new threshold of the confidence level by a SerialEM script redoyoneoLocr without taking the images again. The retry needs to stop yoneoLocrWatch.py running in lowmagxtal mode before starting the job. Please also refer to Fig. 2 for the workflows.

**Reporting summary**. Further information on research design is available in the Nature Research Reporting Summary linked to this article.

## Data availability

All other data in this study are available from the corresponding author upon reasonable request.

## Code availability

The software yoneoLocr including SerialEM scripts and associated programs can be obtained from a GitHub site (https://github.com/YonekuraLab/yoneoLocr). The ParallEM suite is also downloadable from https://github.com/YonekuraLab/ParallEM.

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

## Acknowledgements

We thank D. B. McIntosh for help in improving the manuscript. This work was partly supported by JST-Mirai Program Grant Number JPMJMI20G5 (to K.Y.), JST CREST Grant Number JPMJCR18J2, Japan (to K.Y., S. M.-Y. and K.T.) and the Cyclic Innovation for Clinical Empowerment (CiCLE) from the Japan Agency for Medical Research and Development, AMED (to K.Y.).

## Author contributions

K.Y. designed the project; K.Y. and S.M.-Y. conceived the approach; K.Y. wrote the software; K.T. wrote a python script (convLM2DIFF.py); H.N. setup YOLOv5; S.M.-Y., T.H. and K.T. provided image and diffraction data; T.H., S.M.-Y. and K.Y. tested the software and performance; K.Y. wrote the manuscript; T.H. and K.Y. prepared the tables and movies; All authors discussed the results.

## Competing interests

The authors declare no competing interests.
