## [Peer Review File · Communications Biology]

Reviewers' comments:

Reviewer #1 (Remarks to the Author):

The authors present a multipurpose cryoEM and cryoEX software, yoneoLocr, based on YOLOv5. I am a cryoEM researcher who works alongside deep learning researchers, and I have no practical experience in cryoEX, so I will only comment on the cryoEM and deep learning aspects of the manuscript. The software appears to work well based on the claims in the text. In particular, the targeting of the thick ice holes in Figure 2 is promising. However, proper evaluation metrics are not presented to support these claims. Without these metrics, there's no way the reviewers or readers can objectively evaluate the performance of the software. If the authors properly substantiate their claims, then this work is a welcome step forward in applying deep learning and automation to collection in these fields, which I think will ultimately lead to fully automated workflows. I am glad the authors are working in this fruitful direction, and I would be happy to suggest publication once this issue and one other issue below are addressed.

Comments/concerns/edits are listed below roughly in the order in which they appear in the text. Comments/concerns/edits preceded by a '+' are major and need to be addressed while those preceded by a '-' are minor:

-Line 42-44: In our experience with Legion running on krioses, we can reliably target in holes with an accuracy on the order of 10s of nanometers.

+There are a few other common hole targeting challenges that I think the authors should address to make their software and manuscript more broadly applicable: 1) How well does yoneoLocr work with lacey holes of various shapes and sizes? You can find such data on EMPIAR if required. 2) How does the software work on gold grids, both holey and lacey? 3) Can the software properly detect empty holes so that they may be avoided? These cases would be a great opportunity to test weight re-training in the YOLOv5 architecture, which the authors mention at the end of the main text.

+The authors provide some statistics as percentages throughout the text regarding success rates of their software, but there are no plots/histograms with the raw data nor are there any proper metrics such as average-precision scores or precision-recall curves. Please provide this information. Also, please explain your metrics completely in the Methods.

-Line 296: Directory path typo. There are several typographical issues like this in the Methods.

-It would be a substantial resource and benefit to the field if the authors would release their training images and coordinates to the community along with this manuscript. EMPIAR is able to support the EM data, but I'm not sure about the cryoEX data. Perhaps zenodo can support it if EMPIAR doesn't.

I hope you're all well.
Best wishes,
Alex Noble

Reviewer #2 (Remarks to the Author):

Real-time object locator for cryo-EM data collection by Yonekura et al.

In this work authors present a machine learning-based approach to real time object location using YOLO approach.

1) I do not have any doubt about the technical correctness of the work, however, I have concerns about its potential interest and impact in the cryo-EM community. I do not consider myself an expert on data acquisition in the electron microscope, but I never heard that the most difficult part of this process is the detection of the holes. However, my understanding is that the most challenging and important step is the fast selection of the best possible foil holes as several typical

problems should appear, such as the absence of particles, too thin ice layer (or no ice layer), inhomogeneous ice, and existence of contaminants for example. In this line, in 2016 it was proposed the following paper [1] (not cited in the paper), where authors proposed different metrics to evaluate the quality of foil-holes during acquisition time to improve the throughput and quality of the acquired data (please see Fig 3 in [1]). I like that in the analysis of the cryo-ED data, authors propose the classification and real-time evaluation of the crystals and diffraction patterns, but why did they not do the equivalent for the cryo-EM foil-hole detection? The authors should extend their work in this front proposing real time classification of foil-holes according to their quality, detecting typical problems as the ones presented above to make their contribution appealing and potentially impacting.

[1] J Vargas, E Franken, COS Sorzano, J Gomez-Blanco, R Schoenmakers, AJ Koster, JM Carazo, "Foil-hole and data image quality assessment in 3DEM: Towards high-throughput image acquisition in the electron microscope" JSB 196(3) 515-524 (2016)

2) "In cryo-electron microscopy (cryo-EM) data collection, locating a target object is the most error-prone."

As explained before, I do not agree with this sentence. To this reviewer the most error prone step is determining if the target object is of good quality or not (ice thickness, emptiness, etc).

3) I do not follow this paragraph:

"Modern schemes take 9 - 25 or more image stacks from holes clustered ... Also, during unsupervised data acquisition multiple trials for positioning one hole may be repeated, resulting in possible radiation damage."

4) Line 81 - how many images did you use in the training step? Is the network capable to detect holes at different magnifications when it was trained with a different magnification?

5) Authors present as advantages of using their approach is improved acquisition time and less exposure time which transforms in less radiation time. However, authors do not provide real quantitative proofs of these claims. In a real cryo-EM data acquisition, using your approach do represent any improvement in the 3d reconstruction for example? The same applies to cryo-ED experiments. Authors should think about how to provide quantitative metrics to statistically evaluate the advances of using their approach.

To Reviewer 1

>The authors present a multipurpose cryoEM and cryoEX software, yoneoLocr, based on YOLOv5. I am a cryoEM researcher who works alongside deep learning researchers, and I have no practical experience in cryoEX, so I will only comment on the cryoEM and deep learning aspects of the manuscript. The software appears to work well based on the claims in the text. In particular, the targeting of the thick ice holes in Figure 2 is promising. However, proper evaluation metrics are not presented to support these claims. Without these metrics, there's no way the reviewers or readers can objectively evaluate the performance of the software.

Thank you for the positive comments. We provided Tables 2 and 3 as evaluation matrices for automated SPA and cryo-EX data collection, respectively. With and without running yoneoLocrWatch.py for hole detection, the success rate was increased from 78 to ~100 % (Table 2). The success rate for diffraction data collection was also increased from 52.8% to 100%. Manual selection / registration of good-diffracting crystals usually takes a long time, and our approach can remarkably reduce the time and labor (Table 3). Of course, these are qualitative measures, as the performance of locating holes and crystals and checking diffraction patterns varies depending on each sample condition. We added these explanations in the text.

> -Line 42-44: In our experience with Legimon running on krios, we can reliably target in holes with an accuracy on the order of 10s of nanometers.

I think this is a reasonable threshold. Indeed, we set 300 – 500 nm for SPA data collection with a Krios microscope and SerialEM. We used Legion only once 5 – 6 years ago and do not know how Legion does better for stage alignment.

> +There are a few other common hole targeting challenges that I think the authors should address to make their software and manuscript more broadly applicable: 1) How well does yoneoLocr work with lacey holes of various shapes and sizes? You can find such data on EMPIAR if required. 2) How does the software work on gold grids, both holey and lacey? 3) Can the software properly detect empty holes so that they may be avoided? These cases would be a great opportunity to test weight re-training in the YOLOv5 architecture, which the authors mention at the end of the main text.

Thank you for the suggestion. We added a new figure now Fig. 4. The original weight succeeded in detection of holes recorded at different magnifications ($\times 2.5k$) and holes in gold grids thanks to an improved algorithm in YOLOv5. We could train on lacey holes of various shapes as for small crystals, but we think it is less useful. We developed the program and scheme for automated data collection, particularly, by applying image shifts through deflector coils once the stage is moved to a new position. Please also see the new Fig. 1. Image shift is very accurate and does not need alignment immediately before data taking. This modern scheme is very effective to speed up data collection as written in the text, and essentially needs a regular array of carbon holes. Also the software could be tuned for detection of empty holes and contaminations, but again this slows the data collection down badly. Surrounding holes around one empty

hole may have good ice, and it is not a good way to exclude data from surrounding holes. We added this explanation in the text.

> +The authors provide some statistics as percentages throughout the text regarding success rates of their software, but there are no plots/histograms with the raw data nor are there any proper metrics such as average-precision scores or precision-recall curves. Please provide this information. Also, please explain your metrics completely in the Methods.

Thank you for the suggestion. We added the raw numbers in the text. The metrics like precision scores the reviewer proposed would represent poor positional reproducibility or mechanical instability of the specimen stage, which is not related with the precision of the software. Please also see our replies above. Instead, we think the final success rate with and without the software running would be more proper and so added Tables 2 and 3. Please also see our replies above.

> -Line 296: Directory path typo. There are several typographical issues like this in the Methods.

Corrected to “\”.

> -It would be a substantial resource and benefit to the field if the authors would release their training images and coordinates to the community along with this manuscript. EMPIAR is able to support the EM data, but I'm not sure about the cryoEX data. Perhaps zenodo can support it if EMPIAR doesn't.

Thanks for the suggestion. We plan to upload data to zenodo.

To Reviewer 2

> 1) I do not have any doubt about the technical correctness of the work, however, I have concerns about its potential interest and impact in the cryo-EM community. I do not consider myself an expert on data acquisition in the electron microscope, but I never heard that the most difficult part of this process is the detection of the holes. However, my understanding is that the most challenging and important step is the fast selection of the best possible foil holes as several typical problems should appear, such as the absence of particles, too thin ice layer (or no ice layer), inhomogeneous ice, and existence of contaminants for example. In this line, in 2016 it was proposed the following paper [1] (not cited in the paper), where authors proposed different metrics to evaluate the quality of foil-holes during acquisition time to improve the throughput and quality of the acquired data (please see Fig 3 in [1]). I like that in the analysis of the cryo-ED data, authors propose the classification and real-time evaluation of the crystals and diffraction patterns, but why did they not do the equivalent for the cryo-EM foil-hole detection? The authors should extend their work in this front proposing real time classification of foil-holes according to their quality, detecting typical problems as the ones presented above to make their contribution appealing and potentially impacting.

[1] J Vargas, E Franken, COS Sorzano, J Gomez-Blanco, R Schoenmakers, AJ Koster, JM Carazo, "Foil-hole and data image quality assessment in 3DEM: Towards high-throughput image acquisition in the electron microscope" JSB 196(3) 515-524 (2016)

Thank you for this comment, but we do not check every hole before data collection, as this could slow the data collection down very badly. As described in the text, the modern scheme for SPA data collection uses image shifts to shoot many areas without precheck of those, once the stage is moved to a new position. We usually take 25 image stacks from surrounding holes (please see the new Fig. 1). This can be very effective to speed up data collection as written in the text. Image shift is very accurate and do not need alignment immediately before data taking. Thus, bad images showing thick ice, less particles, and contaminations should be removed after data collection by using other software such as Warp and cryoSPARC Live. It is impractical to skip one hole based on its appearance, as surrounding holes may be good. Thus, we think that the most critical step here is locating the central hole, but we understand that our claim in the original manuscript was a bit strong and we fixed descriptions as,

“locating a target object is error-prone” in Abstract and “This maneuver is error-prone due to discrepancies in stage positions” in Introduction.

Please also see other replies. We added the reference the reviewer mentioned and modified the text as below.

“We could train the detection system with various images such as lacey holes, empty ones, and holes with thick ice and contaminations, as proposed previously²¹, but it would be less useful for automated data collection through image shifts, as this scheme always uses a regular array of holes and skips pre-check of every hole and for speeding the process up. Image selection could be done just after image acquisition by using other software such as Warp¹¹ and cryoSPARC Live²²”.

2) “In cryo-electron microscopy (cryo-EM) data collection, locating a target object is the most error-prone.”

As explained before, I do not agree with this sentence. To this reviewer the most error prone step is determining if the target object is of good quality or not (ice thickness, emptiness, etc).

Please see our replies above. Image shift is very accurate as mentioned, but stage shift is very error-prone due to discrepancies in stage positions between low and intermediate magnifications and a poor positional reproducibility or mechanical instability of the specimen stage. Locating a target object is indeed the most error-prone, but we understand that our claim in the original manuscript was a bit strong and we fixed descriptions as,

“locating a target object is error-prone” in Abstract and “This maneuver is error-prone due to discrepancies in stage positions” in Introduction.

3) I do not follow this paragraph:

“Modern schemes take 9 – 25 or more image stacks from holes clustered ... Also, during unsupervised data acquisition multiple trials for positioning one hole may be repeated, resulting in possible radiation damage.”

Please see our replies above.

4) Line 81 – how many images did you used in the training step? Is the network capable to detect holes at different magnifications when it was trained with a different magnification?

We added this information in Table 1. We added a new figure now Fig. 4. The original weight succeeded in detection of holes recorded at different magnifications ($\times 2.5k$).

5) Authors present as advantages of using their approach is improved acquisition time and less exposure time which transforms in less radiation time. However, authors do not provide real quantitative proofs of these claims. In a real cryo-EM data acquisition, using your approach do represent any improvement in the 3d reconstruction for example? The same applies to cryo-ED experiments. Authors should think about how to provide quantitative metrics to statistically evaluate the advances of using their approach.

We think that 3D reconstructions or atomic models are not suitable demonstrations, as we will probably get similar quality structures from image stacks and diffraction patterns collected without using our software developed here. However, our approaches could reduce waste of time, disk space and human labor as described in the text and new Tables 2 and 3. Please also see our replies above.

Reviewers' comments:

Reviewer #1 (Remarks to the Author):

The authors updated and addressed most of the reviewers suggestions, which is good. In response to reviewer #2: This software is useful in cryoEM for reducing microscope operators' time and potentially increasing microscope throughput and accuracy. The addition of Tables 2 and 3 shows that their software does what is expected; this makes the manuscript stronger. However, I strongly believe that proper model training metrics need to be provided (which is really easy to do!), as I explain below. This will make the manuscript stronger and more palatable to machine learning researchers.

(+ are major suggestions, - are minor)

+In the machine learning literature, proper and widely-accepted training metrics are F1, precision, and recall curves/scores, along with area under the precision-recall curve (AUPRC). These are metrics showing how the model performs during training on the held-out/validation data. YoloV5 saves metrics to the directory `yolov5/runs/train/exp/`. After training, you can find "confusion_matrix.png" and "results.png". These will include precision and recall curves. If you don't have these files, you can make them with `metrics.py` in the `yolov5/utils/` directory. You can then calculate F1 curves, precision-recall curves, and AUPRC.

-Fig 1b: I suggest placing cross-hairs in the center of each red circle. It took me some time to figure out that it is depicting at 4 targets per hole.

Reviewer #2 (Remarks to the Author):

I have an additional comment based on my previous review. From the authors response, I understand that data collection through image shifts is based on acquiring a low magnification image of a foil hole and then capturing high-magnification images of this hole and of all holes close to it (in this paper are 25 holes) by tilting the beam without moving the stage. If this is the case, I understand the authors response and this information should be incorporated to the manuscript. If this is not the case, and some additional low magnification images should be obtained of holes around it, I do not understand the justification of the authors and I still think that they should improve their model to detect low quality holes:

"We could train the detection system with various images such as lacey holes, empty ones, and holes with thick ice and contaminations, as proposed previously 21, but it would be less useful for automated data collection through image shifts, as this scheme always uses a regular array of holes and skips pre-check of every hole and for speeding the process up. Image selection could be done just after image acquisition by using other software such as Warp 11 and cryoSPARC Live 22".

To Reviewer 1

> The authors updated and addressed most of the reviewers suggestions, which is good. In response to reviewer #2: This software is useful in cryoEM for reducing microscope operators' time and potentially increasing microscope throughput and accuracy. The addition of Tables 2 and 3 shows that their software does what is expected; this makes the manuscript stronger. However, I strongly believe that proper model training metrics need to be provided (which is really easy to do!), as I explain below. This will make the manuscript stronger and more palatable to machine learning researchers.

Thank you for the positive comment. Please see our replies below.

> +In the machine learning literature, proper and widely-accepted training metrics are F1, precision, and recall curves/scores, along with area under the precision-recall curve (AUPRC). These are metrics showing how the model performs during training on the held-out/validation data. YoloV5 saves metrics to the directory `yolov5/runs/train/exp/`. After training, you can find “`confusion_matrix.png`” and “`results.png`”. These will include precision and recall curves. If you don't have these files, you can make them with `metrics.py` in the `yolov5/utils/` directory. You can then calculate F1 curves, precision-recall curves, and AUPRC.

Thank you for this useful suggestion. We added “`confusion_matrix.png`” and “`results.png`” in Supplementary Fig. 2 and added the explanation in Methods as,

“... and training metrics in “`confusion_matrix.png`” and “`results.png`” generated through training with YOLOv5 are given in Supplementary Fig. 2. The

confusion matrixes suggest that misassignment may occur for some classes such as “ice” in xtal, “soso” in diff and “edge” in lowmagxtal modes, probably due to small numbers of training objects in the “ice” and “edge” classes and difficult cases in the “soso” class. Of course, the metrics strongly depend on training and validation datasets. Indeed, detection and classification of these objects were correct with the trained weights here in usual cases, but further training should be done with more data and proper labels”.

> -Fig 1b: I suggest placing cross-hairs in the center of each red circle. It took me some time to figure out that it is depicting at 4 targets per hole.

We fixed Fig. 1b with line-drawing of red circles for clarity.

To Reviewer 2

> I have an additional comment based on my previous review. From the authors response, I understand that data collection through image shifts is based on acquiring a low magnification image of a foil hole and then capturing high-magnification images of this hole and of all holes close to it (in this paper are 25 holes) by tilting the beam without moving the stage. If this is the case, I understand the authors response and this information should be incorporated to the manuscript. If this is not the case, and some additional low magnification images should be obtained of holes around it, I do not understand the justification of the authors and I still think that they should improve their model to detect low quality holes:

Thank you for this comment. Yes. We collect high-magnification images from surrounding holes without moving the stage nor acquiring any additional low-magnification images. We fixed the text in Introduction as,

“Modern schemes take 9 – 25 or more image stacks from holes clustered around a central hole (Fig. 1a) and / or even more stacks for one hole (Fig. 1b) through changing deflector coils once the stage is moved to a new position ⁶. No additional image used for alignment is acquired for each surrounding hole”, and in Conclusion as,

“..., as this scheme always uses a regular array of holes and skips pre-check of all surrounding holes around the center one for speeding the whole process up”.

REVIEWERS' COMMENTS:

Reviewer #1 (Remarks to the Author):

The authors have addressed the issues. I believe the article should be accepted.

Best wishes,
Alex Noble

Reviewer #2 (Remarks to the Author):

The authors have answered all my questions accurately and in my opinion the paper is ready to publish.